# Wavefunction Collapse Broadens Molecular Spectrum

**Peter Lebedev-Stepanov** [ORCID]

FSRC "Crystallography and Photonics" RAS, 119333 Moscow, Russia; lebstep.p@crys.ras.ru

**Definition:** Spectral lines in the optical spectra of atoms, molecules, and other quantum systems are characterized by a range of frequencies $\omega$ or a range of wavelengths $\lambda = 2\pi c/\omega$, where $c$ is the speed of light. Such a frequency or wavelength range is called the width of the spectral lines (linewidth). It is influenced by many specific factors. Thermal motion of the molecules results in broadening of the lines as a result of the Doppler effect (thermal broadening) and by their collisions (pressure broadening). The electric fields of neighboring molecules lead to Stark broadening. The linewidth to be considered here is the so-called parametric broadening (PB) of spectral lines in the optical spectrum. PB can be considered the fundamental type of broadening of the electronic vibrational–rotational (rovibronic) transitions in a molecule, which is the direct manifestation of the basic concept of the collapse of a wavefunction that is postulated by the Copenhagen interpretation of quantum mechanics. Thus, that concept appears to be not only valid but is also useful for predicting physically observable phenomena.

**Keywords:** molecular spectrum; electronic-vibrational level; Franck–Condon principle; collapse of wavefunction; spectral line broadening; adiabatic approximation; polymethine dye; linewidth

## 1. Introduction

In the theory of atomic spectra, the important concept of natural linewidth is introduced. Electrons can occupy discrete energy states in the atom. If an electron is in an excited state, it can jump to an energetically lower state by radiating a photon. The result of this is that the lifetime of the excited state is not infinite. In classical theory, the energy of an electron in such a system decays exponentially with time due to it experiencing radiative friction.

A dipole emitter (linear harmonic oscillator) with a frequency $\omega_0$ is determined by the equation

$$\ddot{x} = -\omega_0^2 x - \gamma \dot{x} \tag{1}$$

with a radiative damping coefficient

$$\gamma = \frac{2\omega_0^2}{3c} r_e, \tag{2}$$

where $r_e = \frac{e^2}{4\pi\varepsilon_0 m_e c^2} \approx 2.818 \cdot 10^{-15}$ m is the electron's classical radius and $m_e$ and $e$ are the mass and modulus of the electron's charge, respectively. The solution of (1) is approximately ($\gamma \ll \omega_0$).

$$x = x_0 \exp\left(-\frac{\gamma t}{2} - i\omega_0 t\right). \tag{3}$$

The energy of the oscillator, averaged over one period, decreases exponentially (Figure 1a, smooth black curve)

$$W = \frac{1}{2} m(\dot{x}^2 + \omega_0^2 x^2) = W_0 \exp(-\gamma t). \tag{4}$$

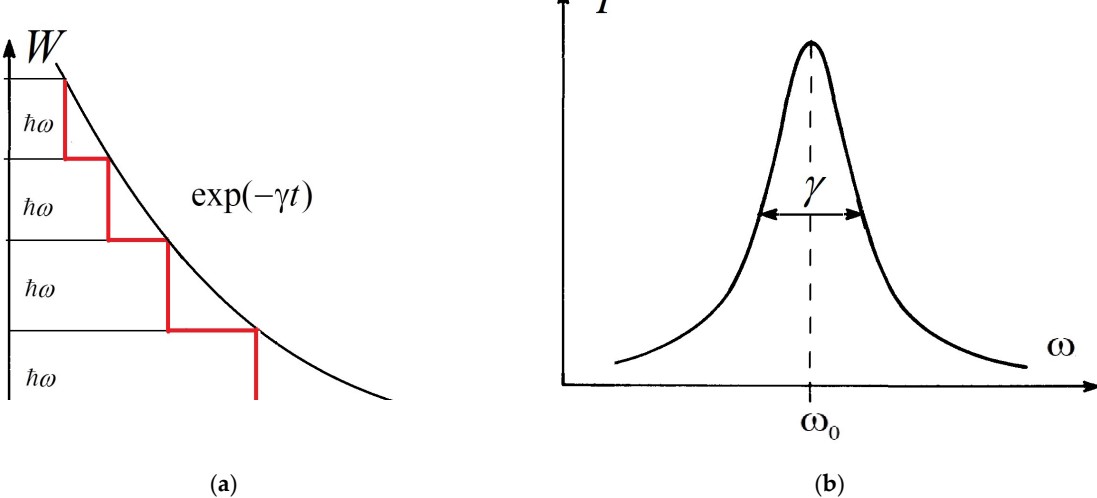

**Figure 1.** Natural linewidth in the theory of atomic spectra: (**a**) in the classical picture, all atoms of an ensemble radiate continuously and simultaneously (black smooth exponential curve); quantum emission of light occurs in portions (red step curve); (**b**) radiation intensity spectral distribution of a classical dipole emitter described by a Lorentzian line; the damping coefficient $\gamma$ determines the characteristic width of the line.

The radiation intensity distribution of this dipole emitter is described by the so-called Lorentzian line (Figure 1b).

$$I = \frac{I_0}{(\omega - \omega_0)^2 + \gamma^2/4}. \tag{5}$$

The damping coefficient $\gamma$ determines the characteristic width at half the height of the Lorentzian, i.e., the breadth at half the maximum intensity distribution. The linewidth introduced here is called the natural linewidth (Ref. [1], pp. 32–33). The atomic decay time (lifetime of excited state) is determined according to $\tau_0 = \gamma^{-1}$. The condition $\gamma \ll \omega_0$ expresses the fact that this time is very long compared with one period of the oscillator $2\pi/\omega_0$. For optical transitions, typical numerical values of $\tau_0$ are $10^{-9} - 10^{-8}$ s, and thus $\gamma = 10^8 - 10^9$ s$^{-1}$ (Ref. [2], p. 308).

Experiments carried out by Wilhelm Wien (1919) confirmed the approximately exponential attenuation of the luminescence of atoms, as well as the order of magnitude of the characteristic value of the coefficient $\gamma$ [3]. However, the frequency dependence of the coefficient $\gamma$ given by Formula (2) has not been experimentally confirmed (Ref. [4], p. 348; Ref. [5], p. 644).

According to quantum theory, an excited atom emits a photon instantly upon transition to a lower energy state. The duration of the emission or absorption process does not appear in the theory and is considered to be negligible. This may seem to be in direct contradiction to Wien's experiments. However, Wien's experiments determined the time of emission not of a single atom but of a large number of atoms simultaneously. For a large number of atoms or molecules, classical and quantum theories lead to qualitatively similar results. According to quantum concepts, each excited atom or molecule has a dark pause, during which the atom is in an excited state but does not radiate. Suddenly, the atom emits a photon instantly.

A quantum picture of the change in the energy of an ensemble of excited atoms with time is shown in Figure 1a (red step curve). That picture differs sharply from the classical one (black smooth exponential curve in Figure 1a). The duration of any dark pause corresponds to the lifetime of the excited state of the individual atom concerned. The lifetimes of atoms even in the same state are different and distributed according to statistical law.

Quantum emission of light occurs in quants. First, one atom emits a photon, then another, and so on. Thus, the curve looks like a staircase. The height of all the steps is the same, but the width of the steps fluctuates chaotically. Each step corresponds to the radiation of an individual atom. However, in the presence of a large ensemble of excited atoms, in practice, it is possible to use an exponential instead of a step curve, as in the classical theory. Thus, if there is a large number of atoms, $N_0$, in the same excited state, then the change in the number of atoms in the excited state is given approximately by an exponential law with decrement $\gamma$:

$$N = N_0 \exp(-\gamma t). \tag{6}$$

The natural linewidth in quantum theory is explained by the uncertainty of the corresponding energy of excited levels of the atom. Heisenberg's uncertainty relation can be written as $\Delta E \tau_0 \geq \hbar$, where $\tau_0 = \gamma^{-1}$. The ground state of the atom has an exact energy value, while the excited state has an energy uncertainty $\Delta E \geq \hbar \gamma$. The typical magnitude of $\gamma$ in conventional spectroscopic units is $10^{-4}$ cm$^{-1}$ (Ref. [2], p. 308), while the optical frequency is of the order of $10^4$ cm$^{-1}$. The general expression for $\gamma$ (Ref. [1], p. 184) is defined as follows:

$$\gamma = \frac{2\pi}{\hbar} \int \rho_W \left| H^2 \right| d\Omega, \tag{7}$$

where $\rho_W$ is the density of energy of the excited atomic states, $\left| H^2 \right|$ is the square of the matrix element of the interaction energy between electron and the electromagnetic field, and $\int d\Omega$ denotes integration over all directions of photon propagation.

Usually, two kinds of spectral-line broadening are considered [6]. Homogenous broadening is due to internal processes that broaden the optical lines in the spectrum of a single atom or molecule (e.g., radiative broadening that provides natural linewidth). Inhomogeneous broadening arises from the effects of the surroundings and external processes. For example, the stochastic electric fields of neighboring molecules lead to Stark broadening that manifests itself in fluctuations of the solvatochromic spectral shift [7,8], while Doppler broadening is due to thermal motion. It is generally accepted that all such broadenings are sufficiently small: many orders of magnitude smaller than the observed width of the optical absorption band of the molecule ($10$–$10^3$ cm$^{-1}$).

There are complex internal processes in molecules that are not present in single atoms. Therefore, molecular spectra, which have the form of bands, do differ significantly from atomic ones. This difference is due to the fact that electronic transitions are influenced by the vibrational-rotational motions of groups of atoms within the molecule. Rovibronic transitions merge into absorption and emission bands corresponding to the energy spectrum of a molecule of this type [6].

An accurate description of quantum transitions in molecules can only be made on the basis of taking into account all the requirements of modern quantum mechanics. Let us recall the basic principles of the Copenhagen interpretation of quantum mechanics (CI), which is taken as a basis by most modern scientists [9–12]:

1. The wavefunction includes complete information about quantum objects and their states (completeness principle).
2. A quantum state represented by the linear superposition of the quantum states can be considered as an admissible quantum state (superposition principle).
3. Quantum objects have certain pairs of complementary properties that cannot all be observed or measured simultaneously (Bohr's principle of complementarity).
4. Heisenberg's uncertainty principle.
5. Max Born's probability interpretation of squared wavefunction.
6. The quantum object under investigation is inseparable from the experimental device used to make the measurements. The interaction between the object and device forms an inseparable part of the quantum phenomena. The instantaneous collapse of the

wavefunction (reduction of the wave packet) upon measurement is a manifestation of that inseparability principle.

7. Quantum and classical physics correspond to each other in the classical limit (Bohr's correspondence principle).

Electronic-vibrational (vibronic) transitions in molecules are usually described on the basis of the Franck Condon (FC) principle. Historically, this principle was introduced in the early years of quantum mechanics (1925–1928) [13–16] before CI was finally formulated.

Therefore, the theory of molecular spectra based on the FC principle does not consistently take into account the postulates of the CI, and, in particular, the FC principle ignores the very important concept of wavefunction collapse. According to Don Howard [16], that is not surprising because the concept of wavefunction collapse was finally formulated only in the mid-1950s by W. Heisenberg: 'Various other physicists and philosophers, including Bohm, Feyerabend, Hanson, and Popper, having further promoted the invention in the service of their own philosophical agendas.'

At the same time, the efficiency of the FC principle in the form in which it came into use in 1928 is proved by its satisfactory agreement with experiments on molecular spectroscopy [17–20]. However, a more consistent application of the postulates of the CI makes it possible to achieve a more detailed description in this area. In fact, the collapse of the wavefunction is manifested in so-called parametric broadening (PB) [21,22] that plays an important role in the formation of molecular vibronic spectra. Herein, consider the mechanism of PB using the example of a (0–0) vibronic transition in a series of polymethine dyes.

Although the electronic-vibrational (vibronic) terms of a molecule shall be considered, the same conclusions can be drawn in the general case of electronic-vibrational-rotational (rovibronic) transitions. Furthermore, although only the absorption spectra shall be considered, the broadening of emission spectra can be determined in a similar way.

The problem of calculating the vibronic transition in the adiabatic approximation was considered in general form in [21,22]. It is also shown there how the collapse of the wavefunction at the moment of absorption or emission of a phonon leads to parametric broadening of the vibronic line. Herein, consider this problem in the simplest one-dimensional case.

## 2. Franck–Condon Principle

In the framework of the adiabatic approximation [6,21–23], the total wave function of an isolated molecule is represented by:

$$\Psi(r,q) = \psi(r,q)\xi(q), \tag{8}$$

where $\psi(r,q)$ and $\xi(q)$ are electronic and nuclear wavefunctions, respectively, and $r$ and $q$ are electronic and nuclear (vibrational) coordinates. The system of adiabatic equations is

$$[\hat{T}_e + U(r,q)]\psi(r,q) = V(q)\psi(r,q), \tag{9}$$

$$[\hat{T}_N + V(q)]\xi(q) = E\xi(q), \tag{10}$$

where $\hat{T}_e$ and $\hat{T}_N$ are the kinetic energy operators of the electrons and nuclei, respectively; $U(r,q)$ is the total potential energy of the molecule. The electronic energy $V(q)$ describes the effective potential energy surface for the nuclei Equation (10).

The probability of a transition between stationary states $\Psi_g \leftrightarrow \Psi_u$, where $g$ denotes the ground electronic state and $u$ the excited state, is proportional to the square of the matrix element of the electric dipole-moment operator

$$M^2 = |\langle n|\mu(q)|n'\rangle|^2, \tag{11}$$

where

$$\mu(q) = \langle \psi_u(r,q) | \hat{\mu} | \psi_g(r,q) \rangle,$$
$$|n\rangle \equiv \xi_g(q), |n'\rangle \equiv \tilde{\xi}_u(q),$$

Within the framework of harmonic approximation, the frequency of absorbed photon $\omega_{un'gn}$ is determined by the energy of a transition from an initial vibrational level $n$ of the ground electronic state to a vibrational level $n'$ of an excited electronic state (Figure 2):

$$\omega_{un'gn} = \hbar^{-1}(E_{un'} - E_{gn}) = \omega_{00} + n'\Omega_u - n\,\Omega_g, \tag{12}$$

where

$$E_{gn} = E_{g0} + n\,\Omega_g \tag{13}$$

and

$$E_{un'} = E_{g0} + \hbar\omega_{00} + n'\hbar\Omega_u \tag{14}$$

are the vibronic energies of the ground and excited states, respectively, $\omega_{00} = \hbar^{-1}(E_{u0} - E_{g0})$ is the frequency that corresponds to a purely electronic transition, while $\Omega_g$ and $\Omega_u$ are the frequencies corresponding to vibrational quants of the ground and excited states, respectively. There are fast motions of electrons and slowly varying nuclear coordinates during electronic-vibrational transition in such a system: $\omega_{00} >> \Omega_i$ ($i = u,g$).

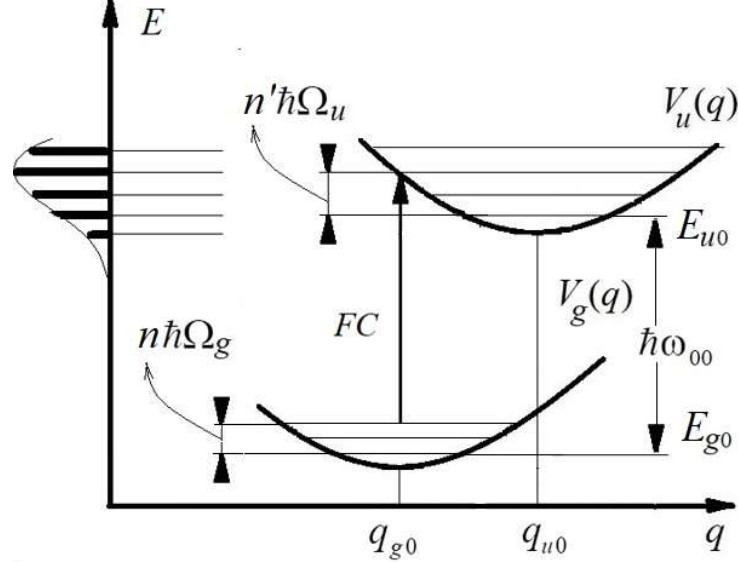

**Figure 2.** Franck–Condon diagram of 'vertical' electronic transitions like *FC* in an isolated molecule between the ground state with nuclear potential energy $V_g(q)$ and the excited (upper) state with potential energy $V_u(q)$. The quasi-continuous spectrum of a series of vibronic transitions located in the vicinity of an *FC* and their envelope that forms a common absorption band are shown on the left. Adapted from Reference [21].

$E_{gn}$ is determined by solving the eigenvalue problem (10), i.e.,

$$[\hat{T}_N + V_g(q)]\xi_n(q) = E_{gn}\xi_n(q), \tag{15}$$

where $V_g(q)$ is the solution of Equation (9), i.e.,

$$[\hat{T}_e + U(r,q)]\psi_g(r,q) = V_g(q)\psi_g(r,q). \tag{16}$$

If $n = 0$ (zero-point vibrations), the squared one-dimensional wavefunction of the linear harmonic oscillator can be written as follows ([24], p. 67):

$$\xi_0^2(q) = \frac{\exp(-(q - q_{g0})^2 R_0^{-2})}{R_0\sqrt{\pi}}.$$ (17)

where the radius of the oscillator is

$$R_0 = \left(\frac{\hbar}{\widetilde{M}\Omega_g}\right)^{1/2},$$ (18)

$\widetilde{M}$ is the reduced mass of nuclei. Formula (17) gives the probability-density distribution of the position of the nucleus in an isolated molecule.

$E_{un'}$ is defined in a similar way.

The oscillator strength of the transition with frequency (12) is given by:

$$f_{un'gn} = n_e\frac{2m_e\omega_{un'gn}}{3e^2\hbar}M^2,$$ (19)

where $m_e$ and $e$ are the mass and electric charge of an electron, respectively, and $n_e$ is the number of electrons which contribute to transition. The spectral intensity is determined by:

$$I_{un'gn}(\omega) = f_{un'gn}\delta(\omega_{un'gn} - \omega),$$ (20)

where $\delta$ is the Dirac delta function.

The adsorption band of a molecule (Figure 2) is an ordered collection of a great number of closely spaced electronic-vibrational terms, $\Psi_{gn} \to \Psi_{un'}$:

$$I(\omega) = \sum_{n,n'=0}^{\infty} P_n I_{un'gn}(\omega),$$ (21)

where $P_n$ are the statistical probability factors averaged over the initial states [6,21,22]. Each term under the sum in Equation (21) is the spectral intensity of the separate vibronic transitions described by the Dirac delta function in accordance with Equation (22), with given numbers $n$ and $n'$. It is important to emphasize that each vibronic line in this model has zero width.

However, in reality, all these lines have finite widths. It has been shown by [21,22] that the collapse of the wavefunction at the moment of emission or absorption of a photon leads to effective broadening of each of the vibronic lines in the spectrum measured using a macroscopic device. Taking that effect into account makes a theoretical description of the band more accurate. This is a problem that will be considered in detail in the next section.

## 3. Concept of Wavefunction Collapse and Modifying the FC Principle

As mentioned in the Introduction, the process of radiation from a molecule has a quantum character. This is expressed by the presence of a more or less prolonged dark phase preceding an almost instantaneous quantum transition. During the dark phase, the molecule does not manifest itself and, as a quantum system, can be considered isolated. Quantum mechanics makes it possible to determine the stationary states in such a quantum system.

There are selection rules that determine the allowed transitions between the stationary levels. However, as long as the molecule does not emit or absorb a photon, it is optically unobservable and does not exhibit any spectral properties. From the standpoint of the CI, this status of the molecule exactly corresponds to a quantum system that exists outside the measurement process and outside the interaction with a classical device. Therefore, the FC diagram of energy levels and transitions (Figure 2) that can be constructed for such an

isolated system is nothing more than an approximate draft of the spectrum because it does not take into account the transition process as it is.

The emission or absorption of a photon by a molecule can be registered by a spectrometer, i.e., a macroscopic device. This is a macroscopic measurement of the quantum system when the molecule loses its isolated status. From the position of CI, this should be accompanied by the collapse of the wavefunction of the quantum object under measurement.

If the collapse of the wavefunction at the moment of emission or absorption of a photon is real, then it has to be manifested in the form of some features of this spectrum. On the other hand, the traditional theoretical description of the spectrum as given in Section 2 does not include such features. If the theory can be improved by taking into account the collapse of the wavefunction, then it will lead to a theoretical prediction of these features in the observable molecular spectrum. Comparison of the theory and experiment should demonstrate their correspondence to each other.

Let us show that taking into account the collapse of the wavefunction actually leads to the prediction of an objectively observable feature of the molecular spectrum that is absent in the conventional approach described in Section 2 of this entry. That feature is the so-called parametric broadening. It was first proposed in [21] and is based on the following consideration.

When a photon is absorbed by a given molecule, i.e., when the molecule interacts with an external electromagnetic field by photon absorption, the photon wavefunction collapses. At this moment, vibronic transition occurs in the molecule. This is the same 'instantaneous' and 'vertical' transition that is postulated by the FC principle.

In the framework of the FC principle and adiabatic approximation, during the transition, the coordinates of the nuclei do not change appreciably. Dramatic changes occur only with the electronic subsystem, and its 'vertical' transition to an excited level of the molecule occurs at the instantaneous position of the nuclei $q$ to that point of the potential energy surface of the excited (upper) state, which is under the corresponding point of the lower curve (potential energy surface of the ground state). It should be emphasized that during photon absorption (or emission), the electronic part of the molecule behaves almost separately and independently from the nuclear subsystem. In this way, the collapse of the wavefunction of the molecule takes place. In this case, the distance along the vertical axis on the FC diagram between the two potential energy surfaces (ground and excited states) depends on the current value of the coordinates of the nuclei in the initial state in which the vibronic transition occurs.

A general consideration of the corresponding vibronic problem in the harmonic approximations was carried out in [21,22]. Let us consider this problem here in a simplified one-dimensional formulation.

Adiabatic Equations (15) and (16) describe the stationary states. The square of the nuclear wave function $\xi^2(q)$ describes the probability-density distribution of position of the nuclei. This makes it possible to determine the average values of energy and other physical quantities describing the system. Therefore, for the average potential and kinetic energies of the nuclei of the initial state, the expressions are

$$\langle V_g \rangle = \int \xi_g(q) V_g(q) \xi_g(q) dq \tag{22}$$

and

$$\langle T_{N,g} \rangle = \int \xi_g(q) \hat{T}_N(q) \xi_g(q) dq \tag{23}$$

respectively. The similar expressions

$$\langle V_u \rangle = \int \xi_u(q) V_u(q) \xi_u(q) dq \tag{24}$$

and

$$\langle T_{N,u} \rangle = \int \xi_u(q) \hat{T}_N(q) \xi_u(q) dq \tag{25}$$

can be written for the final (excited) state with wavefunction $\xi_u(q)$. According to the Equation (15), the eigenvalues of the energies of the initial and the final states are

$$E_g = \langle V_g \rangle + \langle T_{N,g} \rangle \tag{26}$$

and

$$E_u = \langle V_u \rangle + \langle T_{N,u} \rangle, \tag{27}$$

respectively. Note that in the harmonic approximation, the average kinetic energy is equal to the average potential energy that is counted from its minimum ($V_{\min}$):

$$\langle V_g \rangle - V_{g\min} = \langle T_{N,g} \rangle; \quad \langle V_u \rangle - V_{u\min} = \langle T_{N,u} \rangle. \tag{28}$$

During the transition between the initial and final stationary states, the wave function of the nuclei changes. This transition state is not described by stationary adiabatic Equations (15) and (16) and should be considered in a special way.

According to the Franck–Condon principle, the transition between the initial and final stationary states corresponds to definite adiabatic coordinates of the nuclei, $q(t)$, which were caught ('photographed') in the moment $t$ when the photon was absorbed or emitted by the system.

Thus, one can consider some 'instantaneous' intermediate state of the system when the coordinates of the nuclei have some definite values. At the same time, the probability-density distribution is described by the Dirac delta function, $\delta(q - q(t))$, corresponding to the position of the nuclei at the points with coordinates $q(t)$. Such a probability-density distribution corresponds to some 'instantaneous' wave function $\Xi(q)$ of the intermediate state, the square of the modulus of which is equal to the delta function:

$$\Xi^*(q)\Xi(q) = |\Xi(q)|^2 = \delta(q - q(t)). \tag{29}$$

Thus, the collapse of the wavefunction of nuclei is expressed, resulting in the probability-density distribution of the nuclei in the initial state, $\xi_g^2(q)$, being instantly transformed into the Dirac delta function:

$$\xi_g^2(q) \to \delta(q - q(t)), \tag{30}$$

$$\xi_g(q) \to \Xi(q), \tag{31}$$

where $\Xi(q)$ denotes the collapsed wavefunction of the nuclei.

After that, there is a transition from an intermediate nonstationary state with a collapsed wavefunction to the final stationary state with non-collapsed wavefunction:

$$\delta(q - q(t)) \to \xi_u^2(q), \tag{32}$$

$$\Xi(q) \to \xi_u(q). \tag{33}$$

The collapsed wavefunction of the intermediate state $\Xi(q)$ is nonstationary. It satisfies the nonstationary Schrödinger equation describing a singular transition at the moment of collapse of the initial state wavefunction.

During the Franck–Condon transition, the coordinates and kinetic energy of the nuclei do not have time to change. This means that the kinetic energy of the nuclei as well as their coordinates have definite values; namely, the same values that they had in a stationary state at the moment preceding the transition.

Consider the initial stationary state of the nuclei. If the nuclei have the coordinates $q(t)$, then the effective potential energy of the nuclei at this moment is determined by the position of the nuclei on the potential energy surface (Figure 3)

$$V_g = V_g(q(t)). \tag{34}$$

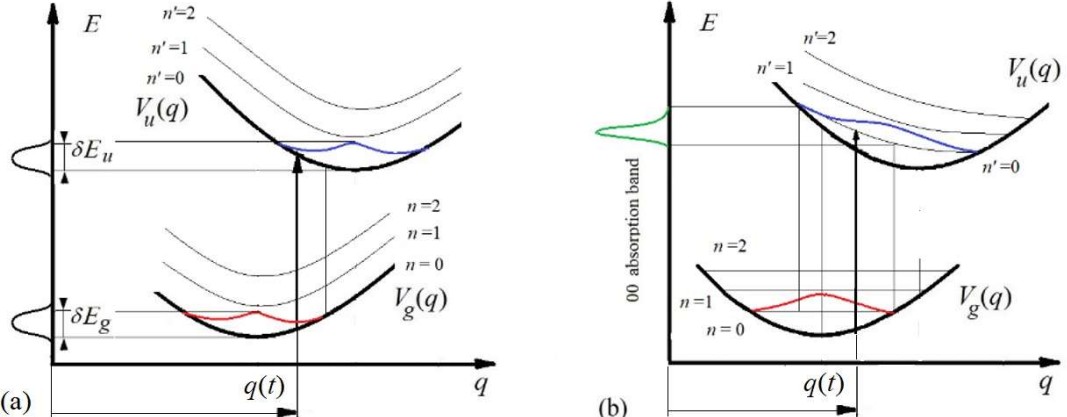

**Figure 3.** A modified Franck–Condon diagrams describing the occurrence of parametric broadening of vibronic levels (**a**) and lines (**b**) in a molecule. Red (and blue) curves indicate the prbability-densities of the nucleus distributions at the ground (and excited) levels as functions of the shift. The green curve describes absorption spectrum corresponding to (0-0) transition, taking into account the statistical weight given by the red curve. Adapted from Reference [21].

Since the total energy of the system in a stationary state is constant and is determined by the expression (26), the instantaneous kinetic energy can be calculated by the formula

$$T_{Ng}(t) = E_g - V_g(q(t)). \tag{35}$$

The conservation of the kinetic energy during the Franck–Condon transition means that in the intermediate state with a collapsed wavefunction, the kinetic energy $T_{Nc}(t)$ is also determined by the Formula (35):

$$T_{Nc}(t) = T_{Ng}(t). \tag{36}$$

At the same time, the potential energy of the intermediate state is not determined, because at this moment there is a spontaneous transition between the surfaces of the potential energies of the initial and final states.

The conservation of the kinetic energy of the nuclei at the moment of transition also means that the kinetic energy in the final state has the same value as it had in the initial and intermediate states:

$$T_{Nu}(t) = T_{Nc}(t) = T_{Ng}(t). \tag{37}$$

The potential energy in the final state at the moment of transition $t$ is determined by the position of the nuclei on the surface of the potential energy (Figure 3):

$$V_u = V_u(q(t)). \tag{38}$$

Then the total energy immediately after the transition of the system to the final state can be represented as

$$E_u(q(t)) = V_u(q(t)) + T_{Nu}(q(t)). \tag{39}$$

The change in energy during the transition determines the frequency of the photon absorbed or emitted by the system

$$\hbar\omega(q(t)) = E_u(q(t)) - E_g = V_u(q(t)) + T_{Nu}(q(t)) - V_g(q(t)) - T_{Ng}(q(t)). \tag{40}$$

However, according to (37), at the moment of transition, the kinetic energies in both states are equal, so the energy of the photon is determined only by the distance between the surfaces of the potential energy corresponding to the coordinates $q(t)$ (Figure 3):

$$\hbar\omega(q(t)) = E_u(q(t)) - E_g = V_u(q(t)) - V_g(q(t)) \equiv \Delta V(q(t)). \tag{41}$$

The coordinates of the nuclei that appear at the moment $t$ have a probability distribution. This probability is determined by the square of the wave function of the stationary initial state.

If ($n$-$n'$) transition is considered, the result is (Figure 3)

$$\omega_{un'gn}(q(t)) = \hbar^{-1}\Delta V(q(t)) + n'\Omega_u - n\,\Omega_g, \tag{42}$$

Taking into account expressions (19) and (20), one can obtain

$$I_{un'gn}(\omega, t) = f_{un'gn}(q(t))\ \delta(\omega_{un'gn}(q(t)) - \omega). \tag{43}$$

The substitution of the integral representation of the delta function in formula (34) gives the following expression:

$$I_{un'gn}(\omega, t) = \frac{f_{un'gn}(q(t))}{2\pi} \int\limits_{-\infty}^{\infty} \exp\left[i(\omega_{un'gn}(q(t)) - \omega)t\right]dt\,. \tag{44}$$

During the dark phase of the molecule, when it does not emit or absorb, the coordinates of the nuclei are characterized by a probability-density distribution of parameter $q$ corresponding to the stationary vibrational wavefunction $\xi(q)$ of the initial state. In the moment of absorption of a photon, the nuclei have certain specific positions $q(t)$, which, in accordance with Formula (44), provide some contribution to the absorbed photon energy. Thus, the energy of the photon can characterize the positions of the nuclei at the moment of absorption of the photon by the molecule. Therefore, registration of the absorption of a photon by the macroscopic device is the equivalent of measurement of the coordinates of the nuclei at the time $t$ of registration. This means that the wavefunction of the molecular nuclei collapses at the time of electron transition $t$ [Formulas (30) and (31)] during the absorption of a photon by the molecule.

The time dependence of Equation (44) has a quasi-stationary, adiabatic character. In other words, it can be described by stationary wave functions with fixed $t$ and given nuclear coordinates $q(t)$ because the nuclei are moving slowly in comparison with the electrons during the electronic-vibrational transition. Using the ergodic hypothesis herein can change the value averaged over $t$ by the same averaged over $q$. Thus, the probability of a given $q$ is determined by the squared nucleus wavefunction of the initial state, $\xi_g(q)^2$ (absorption) or $\xi_u(q)^2$ (emission).

The coordinate $q$ is an adiabatic parameter of the system that determines the instantaneous frequency of electronic-vibrational transition, $\omega_{un'gn}(q)$. Taking into account Equation (44), the spectral density of oscillator strength can be rewritten as:

$$I_{ugnn'}(\omega) = f_{ugnn'}(q(\omega_{ugnn'}))\xi_n{}^2(q(\omega_{ugnn'}))\left|\frac{\partial q(\omega_{ugnn'})}{\partial\omega_{ugnn'}}\right|, \tag{45}$$

where $q(\omega_{ugnn'})$ is the inverse function for the function $\omega_{un'gn}(q)$ given by Formula (33).

Therefore, the average oscillator strength can be determined by the equation:

$$\widetilde{f}_{ugnn'} = \int f_{ugnn'}(q)\xi_n{}^2(q)dq. \tag{46}$$

Let us determine $q(\omega_{ugnn'})$ in the linear approximation. Coordinate $q(t)$ is a current value of the oscillator length in the initial state (Figure 3). This is a ground state if the absorption is considered.

$q(t)$ can be represented as follows

$$q(t) = q_{g0} + \Delta q(t), \quad q_{g0} = \text{const}, \quad q_{g0} >> |\Delta q(t)|. \tag{47}$$

where $\Delta q(t)$ is the time-dependent displacement and $q_{g0}$ is the equilibrium position in ground state (Figure 2).

In the framework of a linear approximation, Equation (42) can be rewritten as

$$\omega_{un'gn}(q(t)) \approx \alpha - \beta \Delta q(t), \tag{48}$$

where

$$\alpha = \frac{\Delta V(q_{g0})}{\hbar} + n'\Omega_u - n\,\Omega_g; \quad \beta = -\frac{1}{\hbar} \frac{\partial \Delta V(q)}{\partial q}\bigg|_{q=q_{g0}}. \tag{49}$$

Hence,

$$\Delta q(\omega_{ugnn'}) = \frac{\alpha - \omega_{ugnn'}}{\beta} \tag{50}$$

and

$$\frac{\partial q(\omega_{ugnn'})}{\partial \omega_{ugnn'}} = -\frac{1}{\beta} \tag{51}$$

Substituting the Formula (51) into the Equation (45), it could be found

$$I_{un'gn}(\omega) = f_{un'gn} F(\omega), \tag{52}$$

where

$$F(\omega) = \left[ \xi_n \left( \frac{\alpha - \omega}{\beta} \right) \right]^2 \frac{1}{|\beta|} \tag{53}$$

is the one-dimensional probability density distribution of $\omega_{un'gn}$:

$$\int_\omega F(\omega) d\omega = 1. \tag{54}$$

For the (0–0) transition (zero-point oscillations), taking into account (17), Equation (44) gives

$$I_{u0g0}(\omega) = \frac{2n_e m_e \omega M_{u0g0}^2}{3\sqrt{\pi} e^2 \hbar |\beta| R_0} \exp\left[ -\left( \frac{\alpha - \omega}{\beta R_0} \right)^2 \right]. \tag{55}$$

The parametrical dependence of the vibronic transition energy on the nuclear coordinate shift $\Delta q$ leads to the slanting vibrational levels in the FC diagram (Figure 3 a,b).

At high temperatures, when non-zero vibrational states of the nuclei are excited significantly, their contribution to the parametric broadening has to be taken into account as well.

The vibronic level of the final state experiences parametric broadening, since the distance between the potential energy surfaces of the ground and excited states depends on the randomly distributed value of the coordinates of the nuclei at the moment of transition (Figure 3). The level of the excited state is broadened in the case of absorption, whereas in the case of emission with transition to the ground state, the energy level of the ground state is effectively broadened.

Thus, the parametric broadening always broadens the vibronic levels and transitions in molecules (including even the ground level), having a fundamental character. This is the reason to include this type of broadening in the category of natural line broadening in molecular spectra. This type of broadening is absent in atoms but, in molecules, usually

exceeds the natural broadening of the lines in atoms, considered in the Introduction, by many orders of magnitude [21,22].

### 4. Estimation of PB in a (0–0) π-Electron Transition

As an example, consider a parametric broadening in the linear polymethine benzothiazole dye with an extended π-electron system (Figure 4a). Experimental optical spectrum of such dye is represented in [25]. Estimations of the PB in absorption spectrum of this dye were performed in [21,22] (Figure 4b) in the framework of a free electrons model derived by H. Kuhn [26]. It was previously shown [22] that the distribution of wavelengths (similar to the frequency distribution) is determined by normal (Gaussian) distribution

$$F(\lambda) = \frac{1}{\sqrt{2\pi}\sigma_\lambda} \exp\left[-\left(\frac{\lambda - \lambda_m}{\sqrt{2}\sigma_\lambda}\right)^2\right], \tag{56}$$

with mathematical expectation (mean value) $\lambda_m \approx 4100$ Å and root-mean-square wavelength deviation $\sigma_\lambda \approx 104$ Å. In this case, the half-width at the half-maximum of the spectrum $I_{00}(\lambda)$ is determined by the formula

$$\delta\lambda_{1/2} = \sigma_\lambda\sqrt{2\ln 2} \approx 123 \text{ Å}. \tag{57}$$

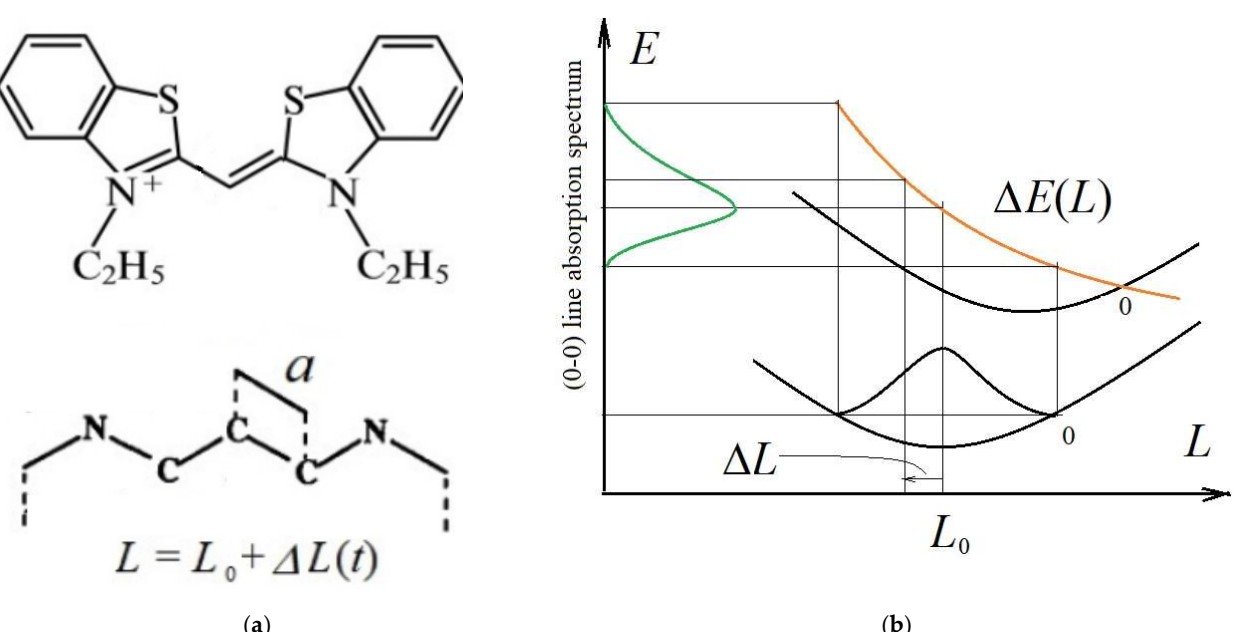

(a)                                                                                                          (b)

**Figure 4.** (**a**) Structure of symmetrical polymethine dye (top); scheme of the π-electron chain of the chromophore in the free-electron model (bottom). (**b**) Nature of the band broadening of (0–0) electronic-vibrational transitions in an extended π-electron system due to the dependence of the vertical FC transition energy $\Delta E$ on the instantaneous length, $L$, of a chain of nuclei, given by equation (60): the modified FC diagram describes the PB of the (0–0) line in the vibronic absorption spectrum (green curve), taking into account the collapse of the nuclear wavefunction in the ground state in the moment of transition; the orange curve denotes the transition energy $\Delta E$ given by Equation (41).

To verify this estimate, a quantum-chemical calculation was performed by the ZINDO/S method to determine the dependence of the wavelength of absorption spectrum maximum of the polymethine dye on the length of interatomic bonds in the central V-shaped C-C-C bridge highlighted in orange in Figure 5a. Figure 5b–d show the modes of normal vibrations in the bridge.

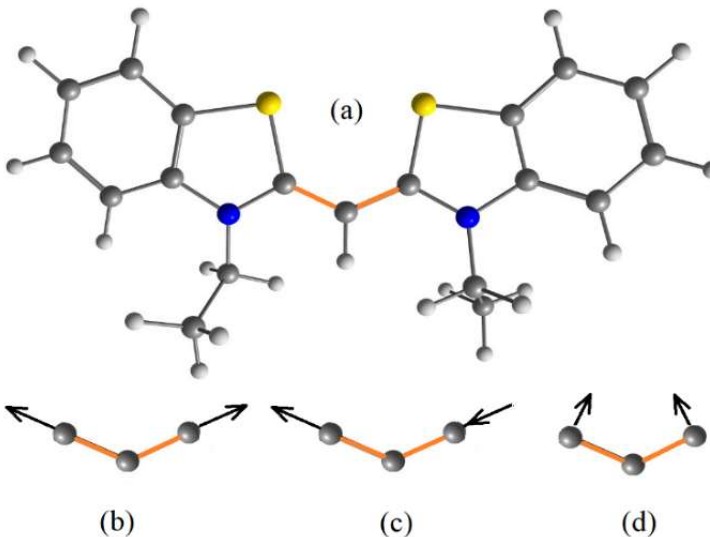

**Figure 5.** (**a**) The polymethine dye corresponding to Figure 4a with V-shaped C-C-C-bridge high-lighted in orange. The modes of normal vibrations of the interatomic bonds in the bridge are represented by (**b**–**d**). Nitrogen atoms are highlighted in blue and sulfur atoms in yellow.

First, the configuration of the molecule was optimized to obtain an equilibrium reference state. Then, a set of C-C bond lengths was selected in the bridge near their equilibrium value ($a_0 \approx 1.40$ Å, Figure 4a) and then wavelengths calculated of the maximum spectral intensity for all bond lengths from the set. Calculations have shown that a noticeable shift in the spectrum occurs only with symmetric oscillations in the bond lengths corresponding to mode Figure 5b, where both C-C bonds in the bridge have the same length at each moment of time. The wavelength that corresponds to equilibrium lengths of the chains is $\lambda_m \approx 4040$ Å approximately (Figure 6).

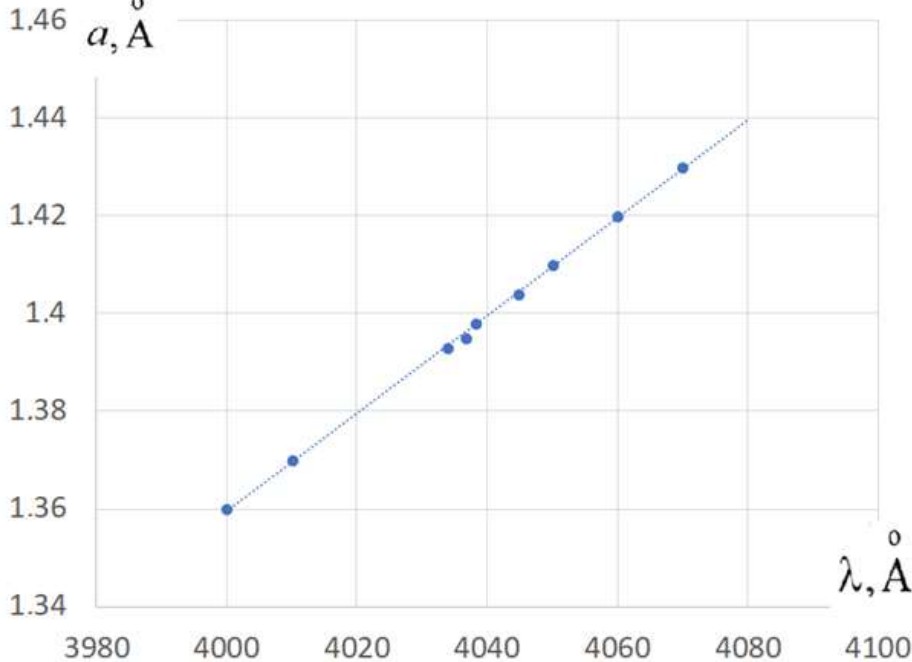

**Figure 6.** Quantum-chemical calculation of PB of the absorption spectrum of the polymethine dye corre-sponding to the Figures 4a and 5a, taking into account only zero-point vibrations in V-shaped C-C-C-bridge. The blue dots in the graph correspond to the calculations by ZINDO/S.

Blue dots in Figure 6 correspond to the set of the values of the C-C bond lengths $a$ for Figure 5b vibration mode and corresponding wavelength $\lambda$ obtained by the computer quantum-chemical calculation. The shift of the maximum value of the spectrum is approximately described by the straight line on the graph in Figure 6: the longer the C-C bond length, the longer the wavelength of the absorbed photon at the maximum value of the spectrum, which is in good agreement with the Kuhn's model. Calculation results can be written as:

$$\Delta a \approx 10^{-3} \Delta \lambda, \tag{58}$$

where $a$ is determined by

$$a = a_0 + \Delta a. \tag{59}$$

The full length of the $\pi$-electron transition $L$ (Figure 4a) is the length of the chain between the nitrogen atoms plus some additional distance on each side [26]

$$L = 6\frac{2}{3}a, \tag{60}$$

According to Peierls [27], when considering a chain of $N$ atoms in the approximation of independent oscillators, the mean square of the uncertainty in the position of the $N$th atom is $N\langle \Delta a^2 \rangle$ that corresponds to the standard deviation of the total length $L$ of this chain of atoms from the mean value. Therefore, from the Equations (59) and (60), it can be formally written that

$$\left\langle \Delta L^2 \right\rangle \approx 6\frac{2}{3} \left\langle \Delta a^2 \right\rangle. \tag{61}$$

These calculations show that only symmetrical length fluctuations contribute to the shift of the spectral maximum wavelength of the (0–0) transition in the framework of a linear approximation. In the considered case, the relationship between the elongation of one C-C bond and the shift of the wavelength of the maximum value of the spectrum is given by the Formula (58).

Therefore, taking into account Formula (58), one can obtain

$$\sqrt{\langle \Delta \lambda^2 \rangle} \approx 10^3 \sqrt{\langle \Delta a^2 \rangle}. \tag{62}$$

For the entire chromophore, the root-mean-square deviation of the wavelength of the spectral maximum from the average value is determined by $\left\langle \Delta L^2 \right\rangle$ given by (61):

$$\sigma_\lambda = \sqrt{\langle \Delta \lambda_\Sigma^2 \rangle} \approx 10^3 \sqrt{6\frac{2}{3}\langle \Delta a^2 \rangle}. \tag{63}$$

The averaging oscillation amplitude of the C-C bond in benzene is $\sqrt{\langle \Delta a^2 \rangle} \approx 0.046$ Å ([28]). Let us use this estimate for calculation in (63):

$$\sigma_\lambda \approx 119 \text{ Å}. \tag{64}$$

Taking into account Figure 6 and Equation (46) with $\lambda_m \approx 4040$ Å, one can obtain the distribution of wavelengths of the (0-0) transition

$$F(\lambda) \propto \exp\left(-\frac{(\lambda - 4040)^2}{2 \cdot 119^2}\right) \tag{65}$$

where wavelength $\lambda$ has to be represented in Å-units. Equation (65) determines the Gaussoid that is shown in Figure 7. Taking into account expression (57), the half-width at the half-maximum of the spectrum is

$$\delta \lambda_{1/2} \approx 119 \cdot \sqrt{2 \ln 2} \approx 140 \text{ Å}. \tag{66}$$

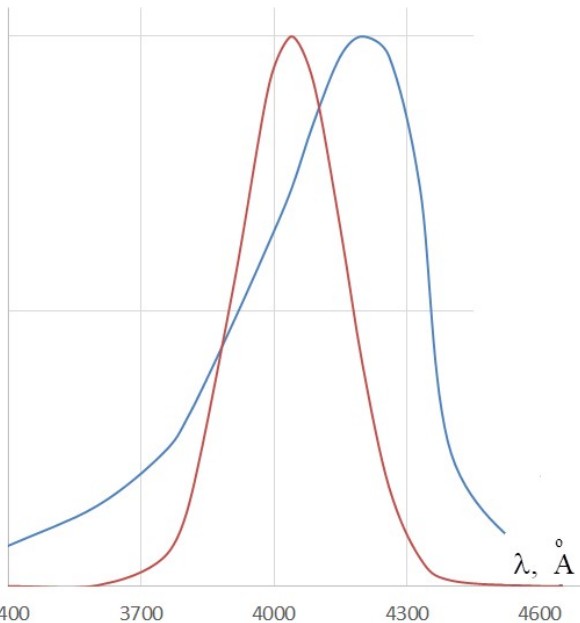

**Figure 7.** Absorption spectra for benzothiazole dye: the red Gaussoid-like curve represents the parametric broadened (0–0) transition line given by Formula (55); the blue curve represents the experimental spectrum of this dye. Adapted with permission from Ref. [25]. Copyright 1940 American Chemical Society.

A similar value obtained from the Kuhn's model, 123 Å, is presented by (57).

Let us compare the obtained broadening with the experimental ones, which can be determined from the spectra given in [25] (see Figure 7). The calculated magnitude of the broadening is smaller than the bandwidth observed in the experiment, but it has the same order of magnitude.

Moreover, the estimation of the PB calculated in the framework of the simple Kuhn's model [22] agrees relatively well with the estimate made using the standard quantum-chemical program ZINDO/S. It should be emphasized that these calculations did not take into account the inhomogeneous broadening associated with the action of solvent molecules to the dye molecule. More detailed quantum chemical calculations would provide better agreement with the experiment.

These calculations illustrate the nature of the PB that occurs due to the collapse of the wave function of a molecule during the emission or absorption of a photon, so the photon energy depends on the configuration of the nuclei at the very short time interval when the electronic transition occurs.

## 5. PB and Single-Molecule Spectroscopy

In classical papers, for example, [25], measurements of the optical spectra were carried out with a large ensemble of identical molecules dissolved in the solution. The number of molecules was large enough to have a sufficiently smooth distribution of molecules over all possible states, taking into account the probability of these. The molecules of the ensemble are therefore simultaneously statistically represented with an arbitrary distribution of the coordinates of the nuclei at the moment of spectrum measurement. Any vibronic line in the spectrum is formed by photons corresponding to the same vibronic transition at a different position of the nuclei at the common moment of measurement. Therefore, statistically weighted contributions for each possible frequency appear simultaneously and in total give a smooth envelope for the spectrum of each vibronic line in accordance with Equation (21).

Exactly the same picture applies to other contributions to the broadening, including those with inhomogeneous broadening of any nature. Inhomogeneous broadening can be minimized by placing the optically active molecules in solid matrices cooled to ultralow

temperatures [29,30]. In this case, the parametric broadening is also minimized, both by increasing the effective rigidity of the vibrational degrees of freedom and by weakening the contribution of non-zero vibrations. For this reason, an experiment showed that the spectrum of a molecular π-electron system placed in a cooled matrix becomes narrower [31].

Over the past 30 years, spectroscopy of single, optically active molecules has achieved great success [32,33]. The measured spectrum of a single molecule differs significantly from the spectrum of a large ensemble of the same type of molecules. The difference is in the high resolution of the spectral lines, which is provided by such spectroscopy of single molecules. In modern experiments with molecular beams, the spectral resolution is limited by the time of flight of the molecules that have a value of about $10^{-3}$ s. This corresponds to a spectral resolution of $10^{-8}$ cm$^{-1}$. In the molecular traps (the cooled, solid matrices noted above), the resolution is even higher. This makes it possible to obtain detailed information about the structure of the electronic and vibrational levels in the molecule. With this in mind, it seems necessary to clarify the concept of spectral linewidth.

The ergodic hypothesis underlying statistical physics declares that the time-average value of a physical quantity characterizing a system is equal to the average statistical value over an ensemble of such systems. Applicable to the case under consideration, this means that the absorption spectrum of an ensemble of identical molecules has to coincide with the absorption spectrum of one (any) molecule from this ensemble accumulated over a sufficiently long time. The same, of course, is true for the luminescence spectrum of the molecule under continuous excitation.

Spectroscopy of a single molecule, which can register a short act of absorption or emission of the vibronic transition of this molecule, registers only a part of the averaged spectrum during each such registration. In this case, the spectrum is provided by photons that correspond to a narrow frequency range and therefore to incomplete statistics. Thus, that measurement of one molecule provides a spectrum with less broadening and a shifted center frequency compared to the spectrum of an ensemble of such molecules. However, taking into account the ergodic hypothesis, if a sufficiently large number of measurements of single molecule are taken, then the averaged spectrum will coincide with the spectrum of an ensemble of the same, non-interacting molecules.

Therefore, while the spectroscopy of a single molecule really only makes it possible to obtain narrowed spectra of the vibronic transitions, the sum of many such measurements coincides in the limit with the spectrum of a large ensemble of the same molecules. This is true for any broadening mechanism, including parametric broadening.

Taking into account the ergodic hypothesis, the width of a parametrically broadened line should be understood to represent a value of the spectrum broadening of the given vibronic transition that is statistically averaged over all possible configurations of the nuclear coordinates, where each configuration corresponds to a particular energy of the vibronic transition. In other words, the linewidth should be understood as the entire effective frequency range that corresponds to a given vibronic transition for a given temperature and external environment. The parametric broadening at zero temperature is described by the standard deviation of the frequency in the spectrum given by Equation (55). To observe the full spectrum, it is necessary to collect a sufficiently large number of statistics, i.e., to register a large number of photons corresponding to the given vibronic transition. For example, this can be achieved by traditional spectrometric measurements of a large ensemble of molecules, as described in [25]. On the contrary, the restricted statistics obtained in the study of a single molecule make it possible to achieve a high resolution for the broadened lines by highlighting only a part of the full spectrum corresponding to each vibronic line.

If the molecules are excited using a laser in a narrow spectral range, then it is obvious that the spectrum of an ensemble of the molecules, as well as the time-averaged spectrum of a single molecule, have to be narrow, since the transitions for that narrow range of frequencies receive a statistical advantage. This method is used in high-resolution laser spectroscopy.

## 6. Conclusions

It has been shown that from the point of view of the Copenhagen interpretation of quantum mechanics, the conventional FC diagram with horizontal levels (Figure 2) describes a molecule as an isolated (unobservable) stationary system, while the proposed bent-level FC diagrams (Figures 3 and 4b) take into account the collapse of the nuclear wavefunction as a result of the FC transition and describe an open system containing the molecule and the measuring device (spectrometer).

In other words, conventional FC diagrams (Figure 2) describing the vibronic transitions depict flat levels that actually occur only for isolated molecules. However, at the moment of transition, the wavefunction of the molecule collapses and so the nuclear and electronic subsystems are broken. In this case, the energy of the emitted photon is determined by the instantaneous position of the nuclei in the initial state at the moment of transition. To take this effect into account, the vibronic levels should repeat the potential energy surfaces of the initial profile (Figure 3a), i.e., in the simplest case, all vibronic levels look curved in the diagram. This leads to the existence of a parametric broadening of each vibronic level and of each vibronic transition in the molecule.

The parametric broadening has the following properties:

1.  PB exists for any electronic-vibrational levels and transitions between them in quantum systems containing two or more interacting atoms or ions, such as a molecule or molecular ion. For individual atoms or ions, this type of broadening is absent.
2.  A non-zero PB also exists for the vibronic level of the ground state. It is important to recall that the radiative broadening of the ground state is equal to zero.
3.  PB belongs to the class of homogeneous broadening because it takes place for a single, isolated molecule with a fixed center of mass at zero temperature.
4.  Since the vibrational and rotational movements of groups of atoms in a molecule depend on the properties of the environment, i.e., neighboring molecule distribution, their temperature, etc., then there is a secondary type of, inhomogeneous, parametric broadening that is a consequence of the surroundings of the molecule.
5.  If zero vibrations of the nuclei predominate, then the line spectrum has a characteristic one-hump Gaussoid-like character (Figure 7). This is seen in the case of low temperatures and where there is high rigidity of the bonds between the atoms in the molecule. If higher (non-zero) vibrational modes have quite significant magnitudes (at higher temperatures), then there are visible contributions within the spectrum corresponding to the presence of two or more spectral maxima.
6.  In a number of specific cases, as, for example, in extended $\pi$-electron systems, the PB has practically the same order of magnitude as the total broadening of the entire band (Figure 7).
7.  Parametric broadening is fundamental since it extends to all vibronic transitions of molecules and it cannot be eliminated. The magnitude of PB actually exceeds the radiative broadening by orders of magnitude, and for the ground state level it is the only type of homogeneous broadening. From this point of view, PB together with radiative broadening, can be considered as the natural broadenings for molecules.
8.  To describe PB, one should use the FC diagrams with bent levels, reflecting the dependence of the potential energy surface on the coordinates of the nuclei at the moment of transition when the nuclear wavefunction collapses.

**Funding:** This research received no external funding.

**Institutional Review Board Statement:** Not applicable.

**Informed Consent Statement:** Not applicable.

**Data Availability Statement:** Not applicable.

**Acknowledgments:** The author thanks Anastasia S. Stepko for quantum-chemical calculations of vibronic transitions in the polymethine dye and Alexandra Ya. Freidzon for a fruitful discussion of the features of generally accepted methods for calculating molecular vibronic spectra.

**Conflicts of Interest:** The author declares no conflict of interest.

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
