# Peer review of "Wavefunction Collapse Broadens Molecular Spectrum"

_encyclopedia, doi:10.3390/encyclopedia3020029_

Round 1

Reviewer 1 Report

The author described the concept of parametric broadening (PB) of electronic-vibrational(-rotational) transitions in molecules, and claimed that it was insofar unaccounted for and resulted from the collapse of nuclear wavefunctions. I found the presentation opaque and the proposed experimental consequences vague (Figure 5), and thus cannot recommend for publication at this stage. I would only reconsider if the authors clearly lay out the difference in experimental predictions between traditional theory and their new concept, and show that the experiments agree (quantitatively) better when PB is taken into account.  

Author Response

Thank you for Review.

  1. Reviewer wrote: “I found the presentation opaque and the proposed experimental consequences vague (Figure 5)”.

I have added to revised manuscript the additional clarified material concluding Figures 6 and 7 (Please see attached file).

  1. Reviewer wrote: “I would only reconsider if the authors clearly lay out the difference in experimental predictions between traditional theory and their new concept, and show that the experiments agree (quantitatively) better when PB is taken into account”.  

Concerning the conventional explanation, unfortunately, the reviewer, maybe, does not know how the broadening is calculated in the existing approaches in quantum chemistry, namely, by the M. Lax method [see, for example, A. Bagaturyants, M. Vener. Multiscale Modeling in Nanophotonics: Materials and Simulation (Pan Stanford Publishing Pte. Ltd, 2018, p. 181]. There is a fitting parameter.  I believe that this parameter must be calculated via parametric broadening.

Also, I recommend to see my previous publication (Lebedev-Stepanov, P. Parametric broadening of the molecular vibronic band due to zero-point oscillations and thermal fluctuations of interatomic bonds. AIP Advances. 2021,11, 035115 (1-13))

https://aip.scitation.org/doi/10.1063/5.0047044

In the attached file, I give a quote from the above-mentioned article, which describes in detail the method used in modern quantum chemistry for calculating organic dye spectra (the M. Lax’ method) and gives its criticism.

  1. Reviewer suggested that manuscript should undergo English revisions. Proofreading and editing of the text were made by a native English speaker. Changes have been made both to the text itself and to the title of the manuscript. The content of the edits made to the Word document can be tracked by activating the red lines on the side of the text of revised manuscript.

Reviewer 2 Report

The article is well-written, informative and useful either for graduate or undergraduate students.

In eq. (3) a minus sign is missing in front of the parameter gamma, I did not check all formulae a double check is in order.

Regarding the plagiarism you detected It cannot be considered such, because the Author refers to his own work, so I would not metion it as a major problem. According to my opinion the text may be left unchanged and proceed with the publication.

Author Response

Thank you for Review.

Reviewer wrote: "In eq. (3) a minus sign is missing in front of the parameter gamma, I did not check all formulae a double check is in order".

That is true correction.  This is my typo. I need also add the dot sign at the top of last x in Eq (1).

Reviewer 3 Report

This paper gives a good review of one aspect of the spectroscopy of molecular vibronic transitions, i.e., parametric broadening. The paper starts with a helpful introduction to atomic and molecular spectra. Then it continues with an extensive review.

I find this paper clear and easy to read. The subject is somewhat outside of my field of research, but I feel the paper is well-suited for the journal. Publish with only minor English checks.

About the plagiarism report:

The major items found by the plagiarism report are mainly the author’s work. I feel it is not unusual to paste fractions of previous papers into a review.    The top fraction seems large to me. Perhaps it would be better for the author to present a short summary of the paper and include a reference, but I feel the narrative is fine way it is.

Author Response

Thanks for the comments.

Reviewer suggested that manuscript should undergo
English revisions.   Proofreading and editing of the text were made by a native English speaker. Changes have been made both to the text itself and to the title of the manuscript. The content of the edits made to the Word document can be tracked by activating the red lines on the side of the text.

Reviewer 4 Report

The author elaborates on “parametric broadening”, a phenomenon which has first been put forward just one year ago by himself in two subsequent publications, and which is based on the idea that if the collapse of the wave function, induced by external measurement, is an objective process, then this process should lead to measurable consequences, in particular, to a broadening of molecular spectral lines. First off, I find the basic idea intriguing, as it puts the premises of the Copenhagen Interpretation (CI) to the test. However, I have serious issues with the actual elaboration on this idea:

  1. The already published papers [21, 22] and the submitted manuscript are basically one and the same article. The text, the calculations, the formulas, the results, the figures, and the tables, are largely identical. Only a somewhat more elaborate introduction, one Figure (Figure 5), and the discussion in section 5 have been added, and some wordings and calculations have been slightly modified. This issue alone is enough to directly reject the paper.
  2. It is not clear to what extent the proposed phenomenon is in agreement with the observations. The spectral broadening observed thus far had readily been explained by conventional analysis (broadening by interaction of superimposed energy levels via electromagnetic field, see e.g. Wigner-Weisskopf model of spontaneous emission), without referring to objective wave function collapse. If the author provides a new analysis based on wave function collapse, then this should resolve some discrepancies between conventional theoretical predictions and observations. Where are these discrepancies? 
  3. The only point where the premise of an objective wave function collapse seems to enter the calculations is in Eq. (33) and (34), where the potential function of ground state and excited state, respectively, are calculated as V_g(Delta q) and V_u(Delta q), with Delta q being the “vector of normal coordinate shifts”, which here means the shift of the instantaneous positions of the nuclei. I have issues with this approach:
    1. The “instantaneous position” of the nuclei is a semiclassical concept resulting from the Born-Oppenheimer approximation. Instantaneous positions of particles are classical, not quantum mechanical, elements introduced into the calculations with the objective to enable feasible calculations. One cannot simply take these semiclassical elements and apply a strictly quantum mechanical concept, the collapse of the wave function, to them.
    2. The collapse of the wave function in this case would rather result in a change of the electronic configuration, and not of the nuclear configuration. It is not clear to me, why the nuclear coordinates q, and not the electronic coordinates r, should change during the collapse.
    3. Ignoring the above issues, and granted that the nuclear coordinates q do change during the collapse, then what does the expressions V_g(Delta q) and V_u(Delta q) actually represent? At this point of the calculation, the author attempts to calculate the variation of the potential energy surface (PES) induced by the collapse. However, such variation should be calculated as Delta V = V(q+Delta q)-V(q), and not as V(Delta q). If V were linear in q, then this would not make a difference, but V is explicitly non-linear, depending quadratically on q. So, his calculations at this critical point, and hence in all that follows, seem false to me.
    4. Thank you for giving me the opportunity to respond to your similarity analysis with regard to possible self-plagiarism in the submitted manuscript. Aside from the algorithm detecting a significant amount of similarity (30%), there is more to say here. Even if the author had rephrased every single sentence, leading the algorithm to detect a much lower degree of similarity, the manuscript would still essentially be a copy of both the original paper [21] and its successor [22]. Not only the wording, but above all the scientific content is largely identical. All three papers propose the same analysis to explain the spectral broadening of molecular vibrational transitions. The results of the numerical calculations are also identical (Table 1 in the manuscript, Table I in [21] and Table 1 in [22]). The submitted manuscript does not contain any new physical content compared to the original work [21] and its successor [22]. The only thing that has changed is a shift in interpretation, namely an increasing confidence in the assignment of the slow fluctuations of the nuclear coordinates (slow compared to the electronic transitions), which are identified as the cause of the broadening of the spectral lines, to the "collapse of the wave function". While in the first paper [21] no mention is made of "collapse of the wave function" and "Copenhagen interpretation", in the second paper [22] we find the controversial statement that the nuclear vibrations are a consequence of a collapse of the core wave function. Now the submitted manuscript makes the same statement and presents the same analysis. While I may disagree with the statement and analysis for reasons set out in my report, particularly because other interpretations are available that do not require an objective collapse of the wave function, the much more critical point is that no new physics is presented. From a physical point of view, everything is already laid out in the first paper [21]. It would be perfectly fine for the author to propose their (controversial) interpretation as a commentary or as an article in a more philosophically oriented journal, but as a contribution in a physical journal it fails the criterion of novelty.

Concluding a cannot recommend the paper for publication.

Author Response

Thanks for the comments. In the following, I will respond point by point to the objections of the reviewer.

  1. Preparing the article at the suggestion of the editors, I proceeded from the fact that the Encyclopedia journal should not present totally new materials, but rather well-tested materials. Therefore, I CONSCIOUSLY used partially previously published, peer-reviewed my own works [21-22] (about 30% of coincidences). At the same time, the article is presented in a new connection, it contains an essentially new consideration related to the analysis of the spectroscopy of a single molecule.
  2. Concerning the conventional explanation, unfortunately, the reviewer does not know how the broadening is calculated in the existing approaches, namely, by the Lax method. There is a fitting parameter that I believe must be calculated via parametric broadening. I discussed this issue with quantum chemists.

Reviewer wrote: “It is not clear to what extent the proposed phenomenon is in agreement with the observations”. The reviewer should read carefully the chapter «5. PB and the single-molecule spectroscopy».

  1. Reviewer wrote: The “instantaneous position” of the nuclei is a semiclassical concept resulting from the Born-Oppenheimer approximation». T The instantaneous position of the nuclei really takes place. This is a strict quantum concept by Max Born related to the interpretation of the wave function as the probability density of the distribution of electrons.

Reviewer wrote: «It is not clear to me, why the nuclear coordinates q, and not the electronic coordinates r, should change during the collapse». This remark of the reviewer indicates that he did not carefully read the text. Let the reviewer show where in the text it says that the electron coordinates do not change. This is nowhere in the text. According to the Franck-Condon principle, during an electronic transition, the electron shell is rearranged completely and instantly, and the nuclei almost do not shift during the transition. This is stated many times in the text when analyzing the Franck-Condon principle. This principle underlies all spectrum calculations. I follow this principle.

Reviewer wrote: «Ignoring the above issues, and granted that the nuclear coordinates q do change during the collapse, then what does the expressions Delta q (Delta q) and V_u(Delta q) actually represent? At this point of the calculation, the author attempts to calculate the variation of the potential energy surface (PES) induced by the collapse. However, such variation should be calculated as Delta V = V(q+Delta q)-V(q), and not as V(Delta q). If V were linear in q, then this would not make a difference, but V is explicitly non-linear, depending quadratically on q». 

Dear reviewer, we consider the small deviations from the minimum of the potential energy surfaces V_g(q_g0) and V_u(q_u0)  , which are determined by Delta q, assuming that the nuclei do not go far beyond the specified region (Figure 3, equation (35)). We expand Delta q (Delta q) and V_u(Delta q) around the minima defined by the coordinates q_g0 and q_u0, as indicated in the text. Therefore, there is no error in the written decomposition. Look, please, carefully, see attached file.

The spatial shift between the positions of the V_g and V_u minima is strictly taken into account in expression (34) using the Delta q_u0k value given by Eq. (37). There is no mathematical incorrectness here. These and other important expressions were earlier rigorously reviewed in articles [21–22]. But I agree that since there was a misunderstanding, it could have been more detailed in the text.

  1. This is the same remark as item 1. I answered it above.

Reviewer 5 Report

The manuscript “Parametric broadening of the molecular vibronic transitions” is devoted to a review and discussion of the causes of spectral line broadening in vibronic spectra. The text of the manuscript is well written and the presentation is very consistent, making it easy for the reader to understand the essence of this work. It was a great pleasure for me to review such a synoptic work dealing with fundamental spectroscopic questions.

Thus, I recommend to accept this manuscript after final proofreading and corrections of some typos (e.g. page 2, line 46: usually when specifying a range of numbers, they are written from lower to higher, so it is better to change 108 - 109 to 10-9-10-8).

Author Response

Thank you for Review.

Reviewer wrote: " I recommend to accept this manuscript after final proofreading and corrections of some typos (e.g. page 2, line 46: usually when specifying a range of numbers, they are written from lower to higher, so it is better to change 108 - 109 to 10-9-10-8)".

That is true correction.  I made this correction in the text.

Round 2

Reviewer 2 Report

The paper is now fine, free of errors and clear

Author Response

Dear Reviewer, thank you for your positive feedback.

Obviously, this positive review does not require new revisions to the manuscript.

Reviewer 5 Report

In my opinion, the changes made by the author in the current version of the manuscript are sufficient.

The results of the similarity rate report do not seem surprising, since this work is in part a summary of a large body of material published previously, but it is also very significant and independent. Especially since most of the similarities relate to the wording of laws, terms, designations, etc., which, of course, could not have been done otherwise. Thus, I believe that this work should be published in its current form.

Author Response

Dear Reviewer, thank you for your positive feedback.

Round 3

Reviewer 2 Report

The present form is suitable for publication

Reviewer 5 Report

Previous comments of the reviewers are taken into account. The article may be accept in its current form.